# To what extent is telehealth reported to be incorporated into undergraduate and postgraduate allied health curricula: A scoping review

Kay Yan Hui☯, Claudia Haines☯, Sophie Bammann☯, Matthew Hallandal☯, Nathan Langone☯, Ciara Williams☯, Maureen McEvoy [ID]☯*

Allied Health and Human Performance, Unit, University of South Australia, Adelaide, Australia

☯ These authors contributed equally to this work.
* maureen.mcevoy@unisa.edu.au

## Abstract

### Background

Telehealth has become a necessity within the medical and allied health professions since the COVID-19 Pandemic generated a rapid uptake worldwide. It is now evident that this health delivery format will remain in use well into the future. However, health education training, most particularly allied health, has been slow to 'catch up' and adapt curriculum to ensure graduates are equipped with the knowledge and skills to implement telehealth in the workplace. The aim of this study was to gain a comprehensive understanding of current telehealth curricula in undergraduate and postgraduate allied health education training programs, with a focus on the aims, objectives, content, format, delivery, timeline and assessments.

### Methods

A systematic search of Medline, Embase, PsychINFO, Scopus, ERIC and relevant grey literature was conducted. Students studying allied health degrees through formal education at either postgraduate or undergraduate level were included, while nursing, dentistry and medical students were excluded. The data from the included studies was extracted and tabulated by country, participants, program and content.

### Results

Of the 4484 studies screened, eleven met the eligibility criteria. All studies were published after 2012, highlighting the recency of research in this area. The studies were conducted in four countries (Australia, United Sates of America, United Kingdom, Norway) and participants were from various allied health professions. Of the included studies, four related to undergraduate programs, four to postgraduate programs and for the remaining three, this was not specified. Curricula were delivered through a combination of online and face-to-

**Data Availability Statement:** All relevant data are within the paper and its Supporting information files.

**Funding:** The author(s) received no specific funding for this work.

**Competing interests:** The authors have declared that no competing interests exist.

face delivery, with assessment tasks, where reported, comprising mainly multiple-choice and written tests.

## Conclusion

Published reporting of telehealth curricula within allied health programs is limited. Even the minority of programs that do include a telehealth component lack a systematic approach. This indicates that further primary research would be beneficial in this area.

## Introduction

Telehealth, defined by the World Health Organization [1] as 'the use of telecommunications and virtual technology to deliver health care outside a traditional health-care facility' is an evolving, yet critical aspect of health care. With the recent COVID-19 pandemic, there has been a dramatic uptake of telehealth by health professionals supported by their governing bodies. In some countries, such as Australia, telehealth was added to the National Medicare Benefits Scheme in May 2020 as the pandemic concern was increasing, allowing patients to access various health services from the safety of their home [2]. In April 2021, as COVID-19 was considered stabilised in Australia, government support for telehealth was extended, indicating recognition of the long-term benefits of the digital delivery health service provision [3]. Similar shifts have been seen in Europe and the UK, where eligibility criteria no longer need to be met to receive subsidised telehealth services [4].

The benefits of telehealth lie in its multifaceted ability to enhance quality of care to various patient demographics [5]. It can be used to effectively perform consultations, assessments, interventions, education, and supervision of patients at home or in remote hospitals [6]. It can also broaden the access of services to disadvantaged populations, such as prisoners and remote patients, limit unnecessary travel time and reduce costs [7,8]. Patient selection and identification is key to determining whether telehealth is a safe and suitable option.

For many allied health professions, it is relatively easy to convert face-to-face consultations to online. This is applicable to physiotherapy, particularly where interventions favour, or progress to, a focus on active management. Research shows that active treatments are more effective at reducing pain, improving motor control, and promoting self-efficacy, in comparison to passive treatments alone [9]. Therefore, allied health services can be tailored to provision by telehealth for appropriate patients. Importantly, patients and caregivers have expressed great satisfaction compared to their standard health care [10].

The recent shift towards telehealth has highlighted a gap in the health system, where professionals may lack the additional skills and knowledge required for telehealth implementation [5]. As a result, we have seen rapid upskilling of professionals via short courses and professional development programs to adapt to the evolving needs of health care [11,12]. This has prompted discussions regarding the inclusion of telehealth delivery skills into curricula, to prepare undergraduate and postgraduate students for the future. Papanagnou et al. summarised the importance of this, stating that "...next generation providers will need to be able to deliver next generation medical care" [5].

There is evolving literature describing how telehealth education is incorporated into medical and nursing degrees. Universities in America have piloted telehealth courses among third-year medical students with some success [13]. Another descriptive study published in 2015 compared the inclusion of telehealth in 43 different nursing programs [14]. A broad review of

the literature into telehealth education across health disciplines and in clinicians, managers, trainees and information technology, undertaken in 2016 by Edirripulige and Armfield [11] included nine studies. Of these, five were in continuing professional development courses and four in formal University courses. A gap remains in allied health curricula in formal University programs, where there is limited evidence of inclusion of telehealth training content. This is surprising, considering the growing demand for this mode of delivery [15].

There are no current recommended guidelines for implementing telehealth into university curriculums [13]. However, a systematic and widespread approach to education and training may hold the key to bridging this gap within the healthcare system. Substantial research, understanding and encouragement may be required before telehealth formulates a core component of all health degrees [13]. Due to the paucity of information available in this area, a scoping, rather than systematic review, is an appropriate study design, with the aim of gaining an overview into what, how, when and where telehealth has been included within allied health training programs globally. This prompted the scoping review question: "To what extent is telehealth reported to be incorporated into undergraduate and postgraduate allied health curricula?"

## Methods

This scoping review used a (PCC) Population, Concept and Context approach as per the University of South Australia Scoping Review Guidelines [16], in line with Joanna Briggs Institute guidelines for scoping reviews [17]. The scoping review protocol was registered on 14/04/2020 titled "To what extent is telehealth reported to be incorporated into undergraduate and postgraduate allied health curricula: A scoping review protocol." This can be accessed via Open Science Framework; registration https://doi.org/10.17605/OSF.IO/ZNYME [18].

### Justification for PCC elements

The population of interest is allied health students. There is currently no standard definition of 'allied health professionals' that is accepted worldwide [19]. Therefore, for the purpose of this scoping review, the 'allied health professions' to be included will be specified (and presented in Table 1 below) based on a list of professions provided by the Australian Government Services Australia [20] and the definition by Health Direct Australia [21]. The concept being explored is telehealth education. Telehealth has been defined as "the use of telecommunication techniques for the purpose of providing telemedicine, medical education, and health education over a distance" [22]. This can be in the form of digital and communication technologies such as computer and telephone. It can be executed in the form of a one to one or group call, with or without video. It also includes technologies used for care such as in-home patient monitoring and storing/transferring data. The context is undergraduate and postgraduate curricula. A 'curriculum', as defined by University of Michigan [23], should include goals for student learning, content, sequence, instructional methods, resources, evaluation processes and assessment. A focus on undergraduate and postgraduate curricula were chosen as this results in some form of recognised or accredited certificate or degree. Curriculum may refer to that of a program or course. In this review we have operationally defined a 'course' as a basic component of an academic program and a 'program' as a combination of courses undertaken during university study to obtain a degree, certificate, or diploma.

### Eligibility criteria

The eligibility (inclusion and exclusion) criteria for the scoping review is detailed below in Table 1.

**Table 1. Eligibility criteria.**

| Studies | Population | Concept | Context |
|---|---|---|---|
| **Inclusion Criteria** | | | |
| Quantitative<br>Qualitative<br>Descriptive<br>Cross Sectional<br>Case Control<br>Case series<br>Cohort<br>Clinical trials<br>Pre-post<br>Quasi-experimental<br>Controlled Clinical Trials<br>Randomised Controlled Trials<br>Systematic Reviews and Literature Review (included in the search only for the purpose of pearling for primary studies)<br>Full text available<br>All dates | Males and females<br>All ages<br>Living anywhere in the world<br>Allied health students of the following professions:<br>• Aboriginal and Torres Strait Islander health practitioners<br>• Audiologists<br>• Chiropractors<br>• Diabetes Educators<br>• Dieticians<br>• Exercise Physiologists<br>• Occupational Therapists<br>• Orthoptists<br>• Osteopathy<br>• Physiotherapists<br>• Podiatrists<br>• Psychologists<br>• Speech Pathologists<br>• Social Workers | Store and transfer of data/ information<br>Video conferencing<br>Patient in home monitoring | Undergraduate degree<br>Postgraduate degree<br>Masters degree<br>Clinical<br>Theory<br>Course curriculum to provide at a minimum: aim, content and structure of course<br>Program curriculum to provide at a minimum: content and structure of courses |
| **Exclusion Criteria** | | | |
| Editorials<br>Opinion Pieces<br>Theses<br>Books<br>Technical Report<br>Non-English studies | Non-human<br>Medicine<br>Nurses<br>Dentists<br>Assistant profession e.g. physiotherapy assistant | Online clinical note taking<br>Skills training for professions using online methods | Professional development courses<br>Exposure to telehealth without prior learning |

## Literature search and information sources

The five databases, Medline, Embase, PsychINFO, Scopus and ERIC, were searched from inception on a date between March 16th 2020 and March 19th 2020. The search was updated in these databases on May 27th 2021 to include studies published in the period of March 2020 to April 30th 2021.

Additional search methods were hand searching relevant journals ('Telemedicine Journal and e-Health' and 'Journal of Telemedicine and Telecare'), Google Scholar search, pearling reference lists of included studies and trawling Government Health and Education Departments. The search for relevant studies and documents published in the period of March 2020 to April 2021 in these journals and grey literature sites, was updated in July 2021.

The key MESH terms used in our Medline search were as follows:

1. Allied Health Personnel/

2. Allied Health Occupations/

3. Occupational therapy/

4. Speech language pathology/

5. Physical Therapy Modalities/

6. Students/

7. Audiologist/

8. Chiropractic/

9. Nutritionists/

10. Orthoptics/

11. Osteopathic Physicians/

12. Podiatry/

13. Social Work/

14. Exercise Physiology/

15. Psychology, Educational/

16. Students, Health Occupations/

17. Telemedicine/

18. Telerehabilitation/

19. Health Education/

20. exp Curriculum/

21. Program Development/

22. Health knowledge, attitudes, practice/

23. Universities/

The full search strategy including keyword search terms conducted in Medline 16/03/2020 can be found in the Supporting Information (S1 Table).

## Study selection processes

The studies were first screened by title and abstract, where words related to "telehealth" and "curriculum" or "education" needed to be included. The screening was undertaken independently and in duplicate by six members of the review team using Covidence. Each study was randomly allocated to different pairs within the review team by Covidence. Prior to the initial screening of the title and abstract, team members independently screened the first 10 studies and compared their decisions regarding eligibility to ensure uniformity in the selection process.

The full-text articles were then obtained for the remaining studies, and the team pairs examined these against the eligibility criteria. The same screening process as used for the title and abstract screening was followed, to ensure consistency. Reasons for exclusions were recorded at this stage of screening. The six members of the review team independently screened the full text of the first five studies and compared decisions. The remaining studies, randomly allocated by Covidence, were then screened independently in pairs. Any disagreements throughout the screening process were discussed and resolved as a group.

Grey literature screening coincided with the search. Potential studies in the grey literature found via title and abstract were screened independently by two reviewers and the results were compared. A search through publications within the last three years of 'Telemedicine Journal and e-Health' and 'Journal of Telemedicine and Telecare' was conducted. The first 100 results from a search of 'Telehealth allied health education' in Google Scholar were also screened. Any

grey literature for which full text screening was undertaken, was also pearled. The grey literature that was deemed relevant by the review team was added to the EndNote library.

## Data collection and extraction process

Relevant data were extracted independently by two members of the research team. A standardised data extraction form was developed by the review team and piloted independently with two included studies by the reviewers undertaking the data extraction. The extracted data were then compared by the two reviewers to check for consistency.

The data items for extraction included the defined allied health professions, organisation and type of degree and how telehealth was defined. The curricula components extracted were the aims/objectives of the course, the content (topics covered), format (lectures, practicals, tutorials), time spent on the topics, mode of delivery (face-to-face, online, blended) and assessment. The same components were extracted for program curricula but the aims/objectives may be replaced by the structure.

## Data synthesis

The data were initially collated and then synthesised in terms of numbers of allied health professional courses/programs where telehealth curricula are taught, the countries, organisation types, level of education (undergraduate or postgraduate programs), aims and objectives, number of hours required, number and structure of courses if related to a program, topics covered, formats of delivery and assessments undertaken. The data were then assembled into meaningful statements.

Two members of the research team reviewed the synthesised data to ensure it reflected the original data. Discrepancies were discussed until mutual agreement was reached.

## Results

### Study selection

The initial search strategy identified 5195 studies and there were two additional studies known to the authors. Following screening and selection, nine studies were included in this review. The updated search on May 27th 2021 identified a further 931 studies. Following screening and selection, two additional studies met the inclusion criteria [24,25]. Therefore, a total of 11 studies were included in this review. The flow of studies through the initial and updated searches is presented in Fig 1.

### Study characteristics

Table 2 provides a summary of the included study characteristics. The eleven studies were published between 2012 and 2021. While most studies were about specific telehealth courses, two of the studies related to telehealth programs [25,26]. The studies were conducted in the following four countries: Australia (n = 4), USA (n = 5), United Kingdom (n = 1), Norway (n = 1). All studies investigated telehealth education among different populations, including undergraduate and postgraduate allied health students. The participants from some studies (n = 6) included both allied health students and students from other programs. In studies where the percentage of allied health students was presented or could be calculated (n = 6), this ranged from 3.1% to 100%. The curriculum reported by Wynn and Ellingsen [25] was terminated in 2018 due to insufficient recruitment of students, while curricula reported by the other ten studies are still on-going.

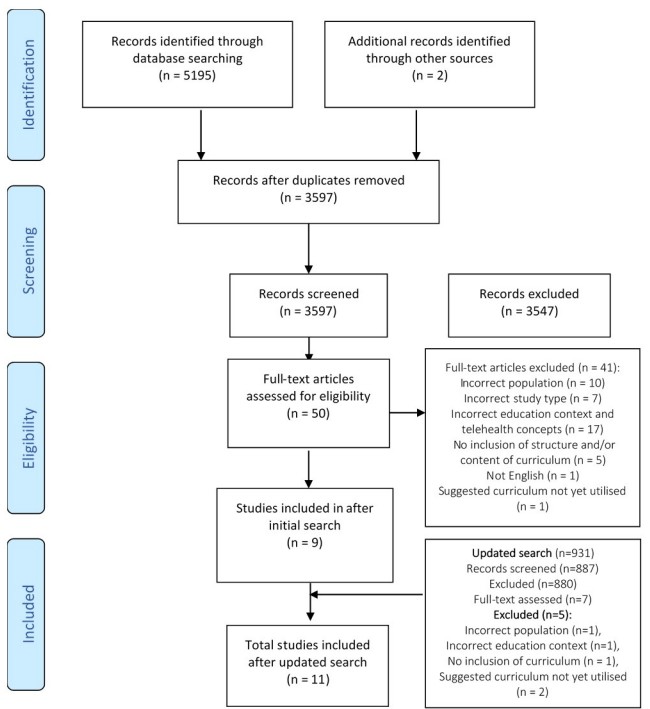

**Fig 1. PRISMA scoping review flowchart.**

## Participant characteristics

The studies included students from both undergraduate and postgraduate programs. There was one study that did not provide the number of participants [34]. In the remaining ten studies, the allied health students varied from 14 [31] to 139 [32]. The allied health students included in the studies by Edirippulige et al. [26], Rutledge et al. [24], Simpson et al. [31], and Serwe and Bowman [34], were from single allied health professions of physiotherapy, social worker, psychology and occupational therapy (OT) respectively. The students enrolled in the course and programs for which curricula were reported in the other seven studies, came from a mixture of programs including the allied health programs of OT, physiotherapy, speech pathology, psychology, audiology, social work, and other non-allied health programs. Overall, in the six studies (Table 2) which reported figures of both total participants and those studying in allied health, 343 of the total 1010 (34.0%) participants were in allied health.

## Characteristics of courses and curricula

Despite all studies investigating telehealth education in university programs, there was variability in how this education was incorporated. Within the eleven studies, three of the curricula courses reported on, related to elective courses [24,27,29], seven were mandatory [25,26,28,31–34], and one study did not specify whether the course was mandatory or elective [30].

The characteristics of the reported telehealth curriculum in the university allied health programs, varied within the included studies. The following tables (Tables 3–6) summarise the curricula aims, content, format, delivery timeline and assessments, which will answer the

**Table 2. Study characteristics.**

| *Author (year) | Design | Country and teaching institution | Participants (n, % of AH students) | Types of participants | Undergraduate/postgraduate program | Mandatory/elective course for AH |
|---|---|---|---|---|---|---|
| Edirippulige 2018 [26] | Retrospective cohort | Australia (The University of Queensland) | N = 32 (1, 3.1%) | PT (n = 1) | Postgraduate (Graduate Certificate, Graduate Diploma, Master) | Mandatory |
| Edirippulige 2017 [27] | Pre-post quasi-experimental | Australia (The University of Queensland) | N = 165 (31, 18.8%) | OT (n = 19), A (n = 5), SP (n = 4), PT (n = 3) | Undergraduate | Elective (Students in health disciplines) |
| Edirippulige 2012 [28] | Cross-sectional | Australia (The University of Queensland) | N = 66 (63, 95.5%) | OT (n = 29), SP (n = 10), PT (n = 20), Psych (n = 4) | Undergraduate | Mandatory |
| Procter 2017 [29] | Cross-sectional | United Kingdom (Sheffield Hallam University) | N = 71 (Unknown n and %) | Unspecified AH students and SW (Unknown n) | Undergraduate | Elective |
| Randall 2016 [30] | Longitudinal pre-/post-test | USA (A Midwestern university) | N = 139 (95, 68.3%) | OT (n = 31), PT (n = 64) | Not specified | Not specified |
| Rutledge 2020 [24] | Prospective cohort | USA (South-eastern University) | N = 67 (Unknown n and %) | SW (Unknown n) | Not specified | Elective |
| Simpson 2014 [31] | Prospective longitudinal | Australia (University of South Australia) | N = 14 (14, 100%) | Psych (n = 14) | Postgraduate (Master) | Mandatory (During the first placement) |
| Sweeney 2018 [32] | Prospective longitudinal | USA (4 Universities at the southeast of America) | N = 594 (139, 23.4%) | SP (n = 49), PT (n = 90) | Not specified | Mandatory |
| Shortridge 2016 [33] | Pre-post quasi-experimental | USA (University of Oklahoma) | N = 137 (Unknown %) | PT, OT (Unknown n) | Undergraduate | Mandatory (3rd year PT and OT) |
| Serwe 2018 [34] | Cross-sectional | USA (Midwestern University) | N = Unknown (Unknown n and %) | OT (Unknown n) | Postgraduate (Master) | Mandatory (In final academic year) |
| Wynn 2020 [25] | Descriptive | Norway (The Arctic University of Norway) | N = 63 (Unknown n and %) | PT, Psych (Unknown n) | Postgraduate (Master) | Mandatory |

*First author name only.

Participants: n = number, AH = allied health, A = audiology, SW = social work, OT = occupational therapy, SP = speech pathology, PT = physiotherapy, Psych = psychology.

question relating to the extent that telehealth is reported to be incorporated into undergraduate and postgraduate allied health curricula.

## Aims and objectives of courses

The studies by Edirippulige et al. [26] and Wynn and Ellingsen [25] which reported on program curricula did not include specific aims. In each of the other studies (n = 9), the courses presented had different aims. The aim was defined as the qualities students were expected to be able to demonstrate on completion of the course. Across the nine studies the three most common course aims were: practice using different technologies in the delivery of health care (n = 6), developing communication skills relevant for telehealth in interprofessional teams or with patients (n = 6), and implementing telehealth in a simulated environment (n = 5). Other aims included understanding basic theoretical and clinical aspects of telehealth (n = 4), evidence-based practice (n = 2) and legal and ethical issues (n = 2). Table 3 provides a summary of the aims and objective.

## Curriculum content in courses or programs

Curricula content varied across the included studies, as summarised in Table 4. The most common reported content (5 or 6 studies) included understanding and practicing the use of

**Table 3. Telehealth course aims and objectives** *.

| Aims/objectives | Edirippulige 2018 [26]** | Edirippulige 2017 [27] | Edirippulige 2012 [28] | Procter 2017 [29] | Randall 2016 [30] | Rutledge 2020 [24] | Simpson 2014 [31] | Sweeney 2018 [32] | Shortridge 2016 [33] | Serwe 2018 [34] | Wynn 2020 [25] ** |
|---|---|---|---|---|---|---|---|---|---|---|---|
| No specified aims of the reported program | ✓ | | | | | | | | | | ✓ |
| Practice using different technologies in the delivery of health care | | | ✓ | ✓ | ✓ | ✓ | ✓ | | ✓ | | |
| Practicing various technologies in a simulated environment | | | | | ✓ | | ✓ | ✓ | ✓ | ✓ | |
| Develop communication skills relevant for telehealth in interprofessional team or with patients | | | ✓ | ✓ | ✓ | ✓ | | ✓ | ✓ | | |
| Understand basic theoretical and clinical aspects of telehealth | | ✓ | ✓ | ✓ | | | | ✓ | | | |
| Evidence-based practice in telehealth | | | | ✓ | ✓ | | | | | | |
| Understand legal and ethical issues associated with telehealth | | | ✓ | ✓ | | | | | | | |

*First author name only;

**only two studies presenting program curriculum rather than a course curriculum.

healthcare technology, utilising case scenarios/role play to practice telehealth and working in an interprofessional team. Less commonly mentioned curricula content (3 or 4 studies) included telehealth research, roles of telehealth, ethical practice, simulated clinics, development of intervention or discharge planning, and setting up telehealth technology. Aspects of content which were least included (1 or 2 studies) in the reported curricula were: adapting clinical competencies for telehealth settings, digital technology use for health information management, home telehealth, telehealth systems evaluation and hospital observation status of telehealth in Australia and changes to telehealth resulting from the COVID-19 epidemic.

## Format of curriculum

While Wynn and Ellingsen [25] listed seven compulsory courses in the first year of the program, the format for delivery was made unclear. The second year was a research thesis [25]. In the remaining ten studies, there were different formats for delivery of the curricula content as summarised in Table 5. Procter [29] was the only study to report the use of a single format being online modules. The remaining ten studies used combined formats, which included lectures, workshops, tutorials, online modules, and delivery of reading material (videos and research papers). Other formats had a self-learning and problem-based components, including presentation, scenario-based discussion, discussion activities, observation and practicals.

The curricula were delivered face-to-face, online or a combination of both. The curricula reported in Simpson et al. [31], and Serwe and Bowman [34] were mainly conducted face-to-face, accompanied by some online reading. The curricula reported in the two studies by

**Table 4. Telehealth curricula content *.**

| Curriculum Content | Edirippulige 2018 [26] | Edirippulige 2017 [27] | Edirippulige 2012 [28] | Procter 2017 [29] | Randall 2016 [30] | Rutledge 2020 [24] | Simpson 2014 [31] | Sweeney 2018 [32] | Shortridge 2016 [33] | Serwe 2018 [34] | Wynn 2020 [25] |
|---|---|---|---|---|---|---|---|---|---|---|---|
| Understand, know and practice using healthcare technology | ✓ | ✓ | | ✓ | | ✓ | | | ✓ | ✓ | |
| Case scenario/role play practice of TH | | | ✓ | | ✓ | | | | ✓ | ✓ | ✓ |
| Practical work in interprofessional teams using TH | ✓ | | | | ✓ | ✓ | ✓ | | ✓ | | |
| Research in TH | ✓ | ✓ | | ✓ | | | | | | | ✓ |
| Roles of TH in epidemiology, population health, public health, health systems | ✓ | ✓ | | ✓ | | | | | | | |
| Ethical TH practice | | | | | | | ✓ | ✓ | | ✓ | |
| Work in simulated clinical situation using TH equipment | | | | | ✓ | | | ✓ | ✓ | | |
| Intervention and discharge planning in TH setting | | | | | | | | ✓ | ✓ | ✓ | |
| Setting up TH technology | | | | | | ✓ | ✓ | ✓ | | | |
| Adapting clinical competencies for TH setting | | | | | ✓ | | ✓ | | | | |
| Use of digital technology in health information management | | ✓ | | | | | | | | | ✓ |
| Observation of how TH is delivered in hospitals | | | ✓ | | | | | | | | |
| TH provision for patients at home | | ✓ | | | | | | | | | |
| TH systems evaluation | ✓ | | | | | | | | | | ✓ |
| Status of adoption of TH in Australia | | ✓ | | | | | | | | | |
| Changes to health and TH resulting from the COVID-19 epidemic | | | | | | ✓ | | | | | |

*First author name only. TH = Telehealth.

Edirippulige and co-authors [26,27] and the study by Procter [29] were delivered via online classes only. All other curricula were a combination of online and face-to-face formats. The curricula outlined by Edirippulige, Samanta and Armfield [27] and Sweeney et al. [32] utilised a program called "Blackboard Inc.", which is an e-learning platform.

Six of the studies had a practical component incorporating patient simulated settings, which was the most common format outlined [28,30–34]. There were only three studies that involved real-life patient placement/internships [30,31,34].

**Table 5.  Telehealth curricula format** *.

| Curriculum Format | Edirippulige 2018 [26] | Edirippulige 2017 [27] | Edirippulige 2012 [28] | Procter 2017 [29] | Randall 2016 [30] | Rutledge 2020 [24] | Simpson 2014 [31] | Sweeney 2018 [32] | Shortridge 2016 [33] | Serwe 2018 [34] | Wynn 2020 [25]** |
|---|---|---|---|---|---|---|---|---|---|---|---|
| Not specified | | | | | | | | | | | ✓ |
| Lectures | ✓ | ✓ | | | | ✓ | | | | ✓ | |
| Workshops | | | | | | | ✓ | | | | |
| Tutorials | | | | | | | | | ✓ | ✓ | |
| Online modules | | | ✓ | ✓ | ✓ | | | ✓ | | | |
| Video material (Online) | ✓ | ✓ | | | | ✓ | | | | | |
| Research paper material (Online) | ✓ | ✓ | | | | ✓ | | | | ✓ | |
| Presentations | | | ✓ | | | ✓ | | | | | |
| Scenario-based discussion (Online) | ✓ | ✓ | | | | | | | | | |
| Discussion activity | | ✓ | | | | | | | | ✓ | |
| Practical in patient simulated settings | | | ✓ | | ✓ | | ✓ | ✓ | ✓ | ✓ | |
| Practical in clinical setting (placement/ internship) | | | | | ✓ | | ✓ | | | ✓ | |
| Observation of clinical teleconsultation | | | ✓ | | | | | | | | |

*First author name only;

** this was a 2 year program with seven courses in first year but format for delivery was not clarified.

## Curricula delivery timeline

Among the eleven studies, all the curricula had different timelines. The shortest one being a one-day practicum with online courses in which the duration was unknown [28]. The longest was a two-year program where the first year was theory based modules and the second year was entirely devoted to a research thesis [25]. The duration of courses, which provided placement/internships, ranged from five weeks in a clinical setting plus three weeks preparation [34] to three semesters [30] with the longest being one year [31]. While the year-long internship included three days of placement each week [31] it was unclear how many placement days were involved for the other two courses [30,34]. Other studies were also vague in the description of the timelines. The curricula reviewed by Shortridge et al. [33] was delivered within three weeks, and that of Sweeney et al. [32] and Rutledge [24] were within two weeks, but the number of days and hours are not specified. Similarly, Edirippulige, Samanta and Armfield [27] did not specify the number of days or hours, but the curricula were delivered over thirteen weeks. The curricula detailed by Procter [29] was delivered as 6 x 10-hour units which were completed one per semester over the duration of the program students were studying.

## Telehealth curricula assessment

Of the eleven studies, four did not state the assessment [28,31,33,34]. The assessments reported for the other seven studies are presented in Table 6. The assessments included multiple-choice

**Table 6. Curricula assessment [*].**

| Curriculum Assessment | Edirippulige 2018 [26] | Edirippulige 2017 [27] | Edirippulige 2012 [28] | Procter 2017 [29] | Randall 2016 [30] | Rutledge 2020 [24] | Simpson 2014 [31] | Sweeney 2018 [32] | Shortridge 2016 [33] | Serwe 2018 [34] | Wynn 2020 [25] |
|---|---|---|---|---|---|---|---|---|---|---|---|
| Not specified | | | ✓ | | | | ✓ | | ✓ | ✓ | |
| Multiple-choice test | | ✓ | | | ✓ | | | | | | |
| Online assessment | | | | ✓ | | | | | | | |
| Reflective writing assignment | | | | | | | | ✓ | | | |
| Scenario-based assessment | | ✓ | | | | | | | | | |
| Presentation | | ✓ | | | | ✓ | | | | | |
| Project—Develop website or app for TH use | | | | | | | | ✓ | | | |
| Research project Dissertation/Thesis | ✓ | | | | | | | | | | ✓ |

[*]First author name only.

test, online assessment, reflective writing assignment, scenario-based assessment, presentation, project and thesis.

## Discussion

This scoping review, which searched the peer reviewed published and grey literature for details of telehealth curricula in undergraduate and postgraduate allied health programs, found the reporting to be sparse. The search to April 30[th] 2021 found eleven studies which were all published between 2012–2020, highlighting the recent evolution of telehealth, and more specifically, the consideration of telehealth training in educational institutions. All studies involved allied health professionals, and the majority also included other health professions. Study types included cross-sectional, prospective longitudinal, pre/post designs (both longitudinal and quasi-experimental), a descriptive study, and a retrospective cohort study.

The results were divided into five key curricula domains: aims/objectives, content, format, delivery timeline and assessment, with variation found across all domains. The aims/objectives and content indicated that generally, most curricula were designed for students to: (a) familiarise themselves with, and practice the use of, technology, (b) practice telehealth in a simulated environment and (c) communicate in interprofessional teams. Most courses used a combination of online and face-to-face formats, undertaken over varying timelines in terms of the length of exposure and the year level of a program when the training was delivered. Assessment of students' knowledge or problem-solving skills was outlined in seven of the eleven curricula, and often involved multiple-choice tests, presentations and theses. Another assessment theme identified was the promotion of self-learning using reflective writing assignments, and projects, along with case study scenario tasks to develop problem solving skills. Several courses offered practicums, placements, or internships; however, the studies did not mention how the students' clinical competency was assessed.

## Curriculum aims and objectives or structure

Aims and objectives are commonly outlined in curricula to provide an indication of the goals and purpose of the training. With the recent introduction of telehealth in healthcare delivery, it may not be surprising that the majority of course objectives beyond exploring the importance and uses of telehealth, involved applied skills. In Edirippulige et al. [26] where the structure of courses in a program was presented rather than the individual course aims/objectives, all courses related to the use of information and communication technologies in clinical practice and included healthcare group communication. The emphasis was on communication skills, simulated experiences and real-life implementation of telehealth in clinical practice, suggesting that telehealth requires new and unique skills in comparison to traditional face-to-face delivery. Wynn and Ellingsen [25] also presented the structure of a two-year Master program. While the second year was dedicated to writing a thesis, in the first year there were seven compulsory courses. The course names were reported but there was no detail of the specific content or delivery format. The content appeared to address theoretical aspects of resources, devices, and methods required for acquiring, storage, retrieval, and use of health information, applications into telemedicine, international health and environmental medicine, research methodologies and consideration of patient and public in telehealth. This program, first offered in 2005 may have been considered an 'early adopter' of innovative ideas, attracting a wide range of international students. Wynn and Ellingsen [25] suggested that the final decline in enrolments may have been due to the loss of the initial 'visionary attraction' as telehealth became more popular, and due to the failure to keep pace with digitization challenges and changes in the health care sector.

## Curriculum content

The most commonly reported curriculum content was learning how to "understand, know and practice using telehealth". This appears to be very general and again may reflect the early development of telehealth curricula. By comparison, the curriculum investigated by Edirippulige, Samanta and Armfield [27] outlined the most comprehensive content with a detailed thirteen-week schedule, which included the practical application of using healthcare technology, research into telehealth and its roles in healthcare systems. The level of detail may be related to the specific aim and design of the study: a pre-post quasi-experimental study measuring the effectiveness of the telehealth course curricula content. Therefore, this objective would entail an expectation of detailed curricula content to be presented as evidence of the intervention. By comparison, Wynn and Ellingsen 2020 [25] provided minimal content detail in their descriptive presentation of an earlier developed, less contemporary program. In general, the focus of a course as reported in a study, also influenced the detail provided on curricula content. For example, Edirippulige et al. [28] explored a narrower area of "student perceptions of a hands-on practicum" with presentation only of relevant practical content. Overall, a trend can be distinguished between the focus of the study and the depth and breadth of presented course content, as differing studies had varying aims, resulting often in tailored, selective reporting of content.

## Curriculum format

Online modules and practical components were common formats of delivery in telehealth teaching. Interestingly, in elective courses there is no practical component with content delivered solely as online modules [29]. This format allows students to complete the modules in their own time between other courses. In line with the aims, some curricula content was weighted heavily with practicals, as in Edirippulige et al. [28], where the benefits of a hands-on

practicum were investigated. In other studies, the format focused on the client experience [34] or the logistics of setting up a remote University clinic [31]. These studies were part of Master programs, where curricula were developed to help drive the implementation of telehealth into clinical practice by placement or internship, rather than simulated setting [31,34].

## Curriculum timeline

In general, curricula were distributed variably depending on the level of study (eg. undergraduate versus postgraduate), degree of content, format (eg. one day intensive practical versus weekly online module) and inclusion of assessments. Not surprisingly, curricula of shorter duration often contained less content and assessment criteria over a smaller period of time. This is shown by Edirippulige et al. [28] who explored two concepts (case scenario practice and observation of telehealth delivery) over a single day practicum, along with online components of an unknown duration in an undergraduate course. In contrast, the curriculum examined by Edirippulige et al. [26] as part of a Master program, encompassed greater depth of content, delivered over an extended period, with more formal assessments. The majority of studies did not state the intensity of curricula timelines, making it difficult to compare the number of hours required to complete each course. For example, courses spanning three semesters may have only consisted of one-hour session per week, in comparison a three-day intensive format.

## Assessment

Assessment in telehealth education courses was the most poorly reported item from data that were extracted. In four of the eleven studies there was no mention of any form of assessment [28,31,33,34], in five studies there was one assessment [24–26,29,30] and in only two studies was there inclusion of more than one assessment throughout the course [27,32]. At a very basic level, the inclusion of any form of assessment was limited. This suggests minimal attention has been paid to this aspect of the curricula at this stage. The construction of quality learning by aligning curriculum, outcomes, teaching and assessment is not a new concept [35]. This may be an area for future development.

## Relevance and recommendations

Telehealth has been used by trained clinicians as a format for allied healthcare delivery for many years now. However, this current review highlights the limited amount of published literature regarding telehealth training within allied health undergraduate and postgraduate programs. In comparison, medicine and nursing students have been exposed to the concepts and skills required to implement telehealth for a longer period [14]. This suggests a general gap within university allied health programs globally.

It was difficult to identify specific trends between the curricula, as each study adopted a slightly different focus. However, in addition to aims/objectives or structure, content and format, all studies stated some form of timeline. This often related to curricula format, and therefore, may indicate that future studies discussing format should also include some form of timeline.

The literature review into telehealth education by Edirippulige and Armfield [11] included four studies where telehealth training was not directed toward continuing professional development in clinicians in the period between 2004–2014. These four studies were therefore potentially eligible for inclusion in the current review. However, only one of these studies [28] met the stricter eligibility criteria applied in the current scoping review, with the others lacking the specifically required detail. The 2016 review found that there was minimal reported detail about telehealth curriculum across health disciplines. However, even with increased rigor, the

current comprehensive review concluded that in published telehealth curricula there are still many gaps.

The 2020 Coronavirus Pandemic has highlighted the importance of telehealth and emphasised that the benefits may extend well beyond times of crises. Interestingly, as recently as 2018, Edirippulige and colleagues stated in a qualitative review, that 'while e-health education is considered important in University medical school curricula, the drivers to support inclusion are not sufficiently strong' [15]. The current scoping review presents an overview of allied telehealth training in educational institutions at a particular "point in time". It will be interesting to see if the current pandemic creates a steep rise in published literature surrounding this topic, especially if governments around the world continue to subsidise the use of telehealth, as seen in Australia, Europe and the UK [4]. Introducing telehealth education at an academic level would help to ensure all allied health professionals receive formal education in the skills needed to implement telehealth in professional practice.

To understand and adapt current curricula to suit evolving needs, further research into current telehealth education and curricula format needs to be undertaken. Ongoing primary research and publication of studies containing detailed curricula could allow for a systematic review to be conducted. Broader areas of telehealth curriculum will need to be addressed in this further research. As noted in this current review, postgraduate telehealth education often focused on implementing telehealth skills into clinical practice, whereas undergraduate courses placed more emphasis on education and simulated skill development. Therefore, comparing the two different approaches to content with the possibility of formulating a single scaffolded curriculum that incorporates both concepts, may be a future direction.

## Strengths and limitations

This review followed the recommended guidelines for conducting Scoping Reviews by the Joanna Briggs Institute [17]. It was completed by a small team of researchers and adopted a rigorous approach. Multiple databases and grey literature were searched, with screening undertaken by researchers both independently and in pairs. Data extraction was undertaken by two independent members of the research team, who piloted a standardised data extraction form prior to extraction to ensure consistency.

However, there are a number of recognised limitations in the scoping review. Firstly, it is acknowledged that while all attempts were made to ensure comprehensive coverage, not all relevant literature may have been identified. While five databases were searched, there are others that may contain additional literature on telehealth curricula not identified in the searched databases. Some grey literature may also have been missed due to the open-ended nature of a scoping review. The exclusion of studies that were not published in English may also have meant that some studies were overlooked. Secondly, there was potential for reporting bias as three included studies were published by Edirippulige et al. [26–28]. Thirdly, while the focus of the systematic review was on curricula for allied health professional students, many studies also included non-allied health students and students from non-health programs. Finally, as this was a scoping review to gain comprehensive coverage of the literature, there was no critical appraisal of the included studies. The quality of the studies and included biases are likely to have varied greatly.

## Conclusions

At this point in time the lack of published literature suggests that there is an education deficit in undergraduate and postgraduate allied health curricula regarding telehealth. There were significant variances and a lack of reporting of telehealth curricula within allied health programs

globally, making it difficult to confidently provide curricula direction. This scoping review forms a basis for universities to consider a structured approach to incorporating telehealth education into their allied health programs. Both academic institutions and healthcare systems would benefit from further primary research into this domain.

## Supporting information

**S1 Fig. PRISMA checklist scoping review.**
(PDF)

**S1 Table. Full search strategy.**
(PDF)

## Author Contributions

**Conceptualization:** Sophie Bammann, Matthew Hallandal, Nathan Langone, Ciara Williams, Maureen McEvoy.

**Data curation:** Kay Yan Hui, Ciara Williams.

**Formal analysis:** Kay Yan Hui, Claudia Haines, Sophie Bammann, Matthew Hallandal, Nathan Langone, Ciara Williams, Maureen McEvoy.

**Investigation:** Kay Yan Hui, Claudia Haines, Sophie Bammann, Nathan Langone, Ciara Williams.

**Methodology:** Kay Yan Hui, Claudia Haines, Sophie Bammann, Matthew Hallandal, Nathan Langone, Ciara Williams, Maureen McEvoy.

**Project administration:** Maureen McEvoy.

**Resources:** Kay Yan Hui, Sophie Bammann, Matthew Hallandal, Nathan Langone, Ciara Williams.

**Supervision:** Maureen McEvoy.

**Validation:** Claudia Haines, Sophie Bammann, Nathan Langone, Maureen McEvoy.

**Writing – original draft:** Kay Yan Hui, Claudia Haines, Sophie Bammann, Matthew Hallandal, Nathan Langone, Ciara Williams, Maureen McEvoy.

**Writing – review & editing:** Kay Yan Hui, Claudia Haines, Sophie Bammann, Matthew Hallandal, Nathan Langone, Ciara Williams, Maureen McEvoy.

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
