## [Decision Letter · Decision Letter 0]

30 Apr 2021

PONE-D-20-35411

To what extent is telehealth reported to be incorporated into undergraduate and postgraduate allied health curricula: A scoping review

PLOS ONE

Dear Dr. McEvoy,

Thank you for submitting your manuscript to PLOS ONE. After careful consideration, we feel that it has merit but does not fully meet PLOS ONE’s publication criteria as it currently stands. Therefore, we invite you to submit a revised version of the manuscript that addresses the points raised during the review process.

The manuscript is interesting and the review process is comprehensive and well described. The only major limitation is the rapid changes in telehealth and in online medical and health professions education brought about by the COVID-19 pandemic. The authors can include more recent studies conducted in this area as suggested by the reviewers. They can try to expand the search duration till 30th April 2021. This will provide a contemporary picture about this important area in medical and health professions education. 

We look forward to receiving your revised manuscript.

Kind regards,

Pathiyil Ravi Shankar

Academic Editor

PLOS ONE

Additional Editor Comments:

Provided earlier

Journal Requirements:

Reviewers' comments:

Reviewer's Responses to Questions

**Comments to the Author**

1. Is the manuscript technically sound, and do the data support the conclusions?

Reviewer #1: Yes

Reviewer #2: Yes

2. Has the statistical analysis been performed appropriately and rigorously? 

Reviewer #1: N/A

Reviewer #2: Yes

3. Have the authors made all data underlying the findings in their manuscript fully available?

Reviewer #1: Yes

Reviewer #2: Yes

4. Is the manuscript presented in an intelligible fashion and written in standard English?

Reviewer #1: Yes

Reviewer #2: Yes

5. Review Comments to the Author

Reviewer #1: While the paper is a well written article on a useful topic, the period chosen for the scoping review has become outdated because of the digital disruption caused by the COVID-19 pandemic. It may be appropriately revised with references to newer publications like: 1. https://www.ncbi.nlm.nih.gov/pmc/articles/PMC7301824/

2. https://journals.healio.com/doi/10.3928/01484834-20200921-06

3. https://www.sciencedirect.com/science/article/abs/pii/S1555415520307261

Reviewer #2: Kindly check the sentence in lines 39 and 40 for correctness.

The current review is performed methodically and reported well, considering that there are no concerns about dual publication and the study is conducted with sound research ethics in place.

6. PLOS authors have the option to publish the peer review history of their article (what does this mean?). If published, this will include your full peer review and any attached files.

Reviewer #1: **Yes: **Suptendra Nath Sarbadhikari

Reviewer #2: **Yes: **Dr Gayatri Ravulaparthi

---

## [Author Response · Author response to Decision Letter 0]

28 Jul 2021

The information presented here has also been included in the rebuttal letter in the uploaded files.

Editor’s comment:

The manuscript is interesting and the review process is comprehensive and well described. The only major limitation is the rapid changes in telehealth and in online medical and health professions education brought about by the COVID-19 pandemic. The authors can include more recent studies conducted in this area as suggested by the reviewers. They can try to expand the search duration till 30th April 2021. This will provide a contemporary picture about this important area in medical and health professions education. 

Authors’ response

Thank you for your comments and recommendations.

The systematic search has been updated as suggested with the search duration expanded to April 30th 2021. As a result two further studies published in 2020 have been included.

Comments to the Author

1. Is the manuscript technically sound, and do the data support the conclusions?

Reviewer #1: Yes

Reviewer #2: Yes

Authors’ response

Thank you

2. Has the statistical analysis been performed appropriately and rigorously? 

Reviewer #1: N/A

Reviewer #2: Yes

Authors’ response

Thank you

3. Have the authors made all data underlying the findings in their manuscript fully available?

Reviewer #1: Yes

Reviewer #2: Yes

Authors’ response

Thank you

4. Is the manuscript presented in an intelligible fashion and written in standard English?

Reviewer #1: Yes

Reviewer #2: Yes

Authors’ response

Thank you

5. Review Comments to the Author

Reviewer #1: While the paper is a well written article on a useful topic, the period chosen for the scoping review has become outdated because of the digital disruption caused by the COVID-19 pandemic. It may be appropriately revised with references to newer publications like: 

1. https://www.ncbi.nlm.nih.gov/pmc/articles/PMC7301824/

2. https://journals.healio.com/doi/10.3928/01484834-20200921-06

3. https://www.sciencedirect.com/science/article/abs/pii/S1555415520307261

Authors’ response

Thank you for the comment. In response, the scoping review has been updated to cover the period to April 30th, 2021.

Thank you also for the suggested additional articles.

Of these, No 2 by Rutledge et al was also found in the updated search and met the inclusion criteria, and has been inclusion in the scoping review.

24. Rutledge C, Hawkins EJ, Bordelon M, Gustin TS. Telehealth Education: An Interprofessional Online Immersion Experience in Response to COVID-19. J Nurs Educ. 2020 Oct 1;59(10):570-576. doi: 10.3928/01484834-20200921-06. PMID: 33002163.

The other two suggested articles did not meet the eligibility criteria. No 1 was not the design for inclusion and No 3 in nurse practitioners was not in the allied health population group to be included in this review.

Reviewer #2: Kindly check the sentence in lines 39 and 40 for correctness.

The current review is performed methodically and reported well, considering that there are no concerns about dual publication and the study is conducted with sound research ethics in place.

Authors’ response

Thank you for the comment. 

The sentence in lines 39 and 40 have been checked for correctness.

Previously the Results section incorporating lines 39 and 40 read:

Of the 3595 studies screened, nine met the eligibility criteria. All studies were published after 2012, highlighting the recency of research in this area. Many were Australian studies (44%), and participants were from various allied health professions. Of the included studies, four related to undergraduate programs, three to postgraduate programs and for and two this was not specified. Curricula were delivered through a combination of online and face-to-face delivery, with assessment tasks, where reported, comprising mainly multiple choice and written tests.

With the updated results included and re-wording for correctness this now reads:

Of the 4484 studies screened, eleven met the eligibility criteria. All studies were published after 2012, highlighting the recency of research in this area. The studies were conducted in four countries (Australia, United Sates of America, United Kingdom, Norway), and participants were from various allied health professions. Of the included studies, four related to undergraduate programs, four to postgraduate programs and for the remaining three, this was not specified. Curricula were delivered through a combination of online and face-to-face delivery, with assessment tasks, where reported, comprising mainly multiple choice and written tests.

6. PLOS authors have the option to publish the peer review history of their article (what does this mean?). If published, this will include your full peer review and any attached files.

Do you want your identity to be public for this peer review? For information about this choice, including consent withdrawal, please see our Privacy Policy.

Reviewer #1: Yes: Suptendra Nath Sarbadhikari

Reviewer #2: Yes: Dr Gayatri Ravulaparthi

Additional changes to the manuscript 

In line with the updating of the review, with the search duration expanded to April 30th 2021, there were two additional studies included in the review. This has resulted in minor changes eg reference numbers have been changed in response to these additional studies and data relating to these two studies have included in Tables. There have also been appropriate changes throughout in the Abstract, body of the Methods, Results, and Discussion sections (red track changes), to account for the additional studies and to consider the changes in this topic area in the period since the original manuscript was submitted. 

Authors’comment

Thank you to the reviewers for your contribution to this review process.

---

## [Decision Letter · Decision Letter 1]

9 Aug 2021

To what extent is telehealth reported to be incorporated into undergraduate and postgraduate allied health curricula: A scoping review

PONE-D-20-35411R1

Dear Dr. McEvoy,

We’re pleased to inform you that your manuscript has been judged scientifically suitable for publication and will be formally accepted for publication once it meets all outstanding technical requirements.

Kind regards,

Pathiyil Ravi Shankar

Academic Editor

PLOS ONE

Additional Editor Comments (optional):

Reviewers' comments:

Reviewer's Responses to Questions

**Comments to the Author**

1. If the authors have adequately addressed your comments raised in a previous round of review and you feel that this manuscript is now acceptable for publication, you may indicate that here to bypass the “Comments to the Author” section, enter your conflict of interest statement in the “Confidential to Editor” section, and submit your "Accept" recommendation.

Reviewer #2: All comments have been addressed

2. Is the manuscript technically sound, and do the data support the conclusions?

Reviewer #2: Yes

3. Has the statistical analysis been performed appropriately and rigorously? 

Reviewer #2: Yes

4. Have the authors made all data underlying the findings in their manuscript fully available?

Reviewer #2: Yes

5. Is the manuscript presented in an intelligible fashion and written in standard English?

Reviewer #2: Yes

6. Review Comments to the Author

Reviewer #2: (No Response)

7. PLOS authors have the option to publish the peer review history of their article (what does this mean?). If published, this will include your full peer review and any attached files.

Reviewer #2: **Yes: **Gayatri Ravulaparthi

---

## [Editor Report · Acceptance letter]

11 Aug 2021

PONE-D-20-35411R1 

To what extent is telehealth reported to be incorporated into undergraduate and postgraduate allied health curricula: A scoping review 

Dear Dr. McEvoy:

I'm pleased to inform you that your manuscript has been deemed suitable for publication in PLOS ONE. Congratulations! Your manuscript is now with our production department. 

Kind regards, 

on behalf of

Dr. Pathiyil Ravi Shankar 

Academic Editor

PLOS ONE